# Socio-Economic Risks Posed by a New Plant Disease in the Mediterranean Basin

**Gianluigi Cardone [1], Michele Digiaro [1], Khaled Djelouah [1], Michel Frem [1,2,3,*], Cosimo Rota [1] , Alessia Lenders [4] and Vincenzo Fucilli [3]**

1 Mediterranean Agronomic Institute, CIHEAM BARI, Via Ceglie 9, Valenzano, 70010 Bari, Italy
2 Lebanese Agricultural Research Institute, El Roumieh Zone, Qleiat, Keserwan, Lebanon
3 Department of Agro-Environmental and Territorial Sciences, University of Bari—Aldo Moro, Via Amendola 165/A, 70126 Bari, Italy
4 Sustainable Land Management, SLM Partners Ltd., Eardley House 182–184 Campden Hill Road, London W8 7AS, UK
* Correspondence: mefrem@lari.gov.lb

**Abstract:** *Xylella fastidiosa* (Wells 1987, hereafter *Xf*), the causal agent of several devastating plant diseases, is threatening new countries of the Euro-Mediterranean, Balkans, Middle East, and North Africa (MENA) regions. In this perspective, a study was carried out to: (a) explore the potential establishment and spread and losses caused by *Xf* in Euro-Mediterranean countries (i.e., France, Greece, Italy, Portugal, and Spain) and the Balkans (i.e., Albania, Bosnia and Herzegovina, Croatia, Montenegro, North Macedonia, Serbia, and Slovenia); (b) assess the potential introduction of *Xf* in the MENA countries (i.e., Algeria, Egypt, Israel, Jordan, Lebanon, Libya, Morocco, Palestine, Syria, Tunisia, and Turkey); and (c) project the socio-economic impacts of *Xf* on olives, grapes, citrus, and almonds in these countries. A novel socio-economic risk assessment technique was developed and applied for these purposes. It revealed that Albania had the highest risk for *Xf* dispersal. In addition, the risk assessment also confirmed the vulnerability of Euro-Mediterranean countries in terms of *Xf* dispersal. In the MENA and Balkans regions, countries with fragmented and small farms are likely to face the worst social impacts, whereas the Euro-Mediterranean region runs the highest economic losses on the target crops.

**Keywords:** biological invasion; economic costs; land use changes; pathways and vectors; pest risk assessments; socio-economic pest impacts; spatial analysis; *Xylella fastidiosa*

## 1. Introduction

International trade appears to be largely involved in the spread of plant diseases through agricultural imports of live plants, forest products, seeds, fruits, and vegetables. Imports arrive from areas of origin, where pathogens are present, to importing countries where the environmental conditions and climate may be suitable for their dispersal [1,2]. However, infected plants may also remain asymptomatic in the early stage of the infection and consequently induce severe economic damages [3]. *Xylella fastidiosa* (hereafter *Xf*), a xylem-limited bacterium, is considered an example of this phenomenon [4]. Indeed, this pathogen has been intercepted and has been already declared in several European and Mediterranean countries [4–12]. These recent outbreaks and interceptions of imported infected horticultural plants and nursery stock show that *Xf* may enter new countries via the global trade network [13–15].

Similarly, the entry and spread of invasive alien organisms are increased by the numerous passengers in international air travel carrying potentially infected plant material [16], which easily escapes airport inspections and weak phytosanitary regulations. Maritime transport also plays a crucial role in the potential movement of alien species. Another risk

factor for *Xf* entry is represented by ships that may transport infected insect vectors as hitchhikers.

In this way, *Xf* may overcome another geographical obstacle, exposing the Euro-Mediterranean (i.e., France, Greece, Italy, Portugal, and Spain), the Balkans (i.e., Albania, Bosnia and Herzegovina, Croatia, Montenegro, North Macedonia, Serbia, and Slovenia), and the Middle East and North Africa (MENA) (i.e., Algeria, Egypt, Israel, Jordan, Lebanon, Libya, Morocco, Palestine, Syria, Tunisia, and Turkey) countries to different levels of introduction risk [17]. As a result of its establishment and spread, *Xf* can cause unacceptable socio-economic [18] and ecological impacts [19,20] as well as public and private management costs [21,22].

*Xf* settles in the xylem, where it can move downstream or upstream [23,24] in more than 563 plant host species [14], including olive, grapes, citrus, and almond fruits. Besides these vulnerable economic crops, *Xf* has been detected in several alternative hosts (i.e., ornamental plants, shrubs, and forest trees) under natural conditions, such as *Polygala myrtifolia*, *Nerium oleander*, and *Prunus avium*. Upon infection, *Xf* is known to cause many different diseases such as Pierce's disease in grapes in California [4], citrus variegated chlorosis in Brazil [25], oak leaf scorch in Florida [26], oleander leaf scorch [27], coffee leaf scorch [28], olive quick decline syndrome in Italy [5]. However, infected plants may also remain asymptomatic without causing any serious damage in the early stage of the infection.

Therefore, there is a high risk that *Xf* will continue to spread to new countries and regions through the movement of infected host plants (asymptomatic or unknown hosts) or through the unintentional transport of insect vectors on goods or vehicles. Moreover, a simultaneous wide geographical scale assessment on *Xf* has never been carried out. In addition, the global socio-economic research on *Xf* is very scarce compared to technical and managerial aspects of this epidemic [29–34]. Furthermore, if studies on the potential socio-economic impact of *Xf* in Europe are scarce, those of *Xf* in MENA countries are even more lacking. The lack of socio-economic impacts of *Xf* at this scale provides some context to this research. By quantifying the losses of *Xf* in the target crops with the best available scientific evidence, this study contributes to covering this gap and enriching the scientific literature on the economic impact in the analysis of pest risk such as *Xf*, mainly in the Euro-Mediterranean basin. Due to the high spread potential and high risk of economic and ecological impacts posed by *Xf*, there is a need for an interregional-level risk assessment for *Xf* simultaneously among Euro-Mediterranean countries, the Balkans and the MENA regions, allowing the selection of priority countries for the potential allocation of financial resources by potential financial donors. This includes an estimate of the socio-economic impacts of *Xf* on the main vulnerable economic crops such as olives (*Olea europaea)*, grapes (*Vitis vinifera)*, almonds (*Prunus dulcis)*, and *Citrus* spp. As such, in this study we assess the likelihood of spread of *Xf* and associated losses caused by this bacterium in the European Mediterranean region and the Balkans and the introduction of *Xf* in the MENA region.

For this purpose, a potential epidemic effect of the pathogen was simulated based on the risk of establishment and spread of the bacterium in target countries taking into consideration the EFSA "*Update of the Scientific Opinion on the risks to plant health posed by Xylella fastidiosa in the EU territory*" [35], "*Potential socio-economic impact of Xylella fastidiosa in the Near East and North Africa (NENA): Risk of introduction and spread, risk perception and socio-economic effects*" [18]. This paper provides a useful referential tool that can also be applied to other pests and areas to increase awareness and support policymakers and other stakeholders in taking appropriate preventive management measures against invasive alien species, especially in noninfected countries.

## 2. Materials and Methods

In this paper, we divided the assessment approach into two major steps: (i) the assessment of the risk of *Xf*'s potential establishment and spread on its main host crops, and (ii) the estimation of the potential socio-economic impacts of *Xf* in Euro-Mediterranean

countries, in the Balkans, and in the MENA region. As such, this section is structured into three parts as follows: (i) data compilation, (ii) an estimation method for the risk of *Xf*'s potential establishment and spread, and (iii) an estimation method for the potential socio-economic impacts of *Xf*.

### 2.1. Data Compilation

For the assessment of the risk of *Xf*'s potential establishment and spread on the main crops, data and information were related to the following issues:

1. Existence of a surveillance program against the establishment and spread of *Xf*;
2. Presence of potential vectors of *Xf*;
3. Climate suitability to the establishment and spread of *Xf*;
4. Abundance of alternative hosts;
5. Abundance of the main crops (olives, grapevines, *Citrus* spp., and almonds) in agricultural land;
6. Availability of national programs for the certification of the plant propagation material of the main crops.

For issues 3, 4, and 5, data and information were obtained directly from official public sources (Supplementary Material S1). Issues 1, 2, and 6 were analysed based on answers obtained directly by the officers of phytosanitary services and/or quarantine services of the respective countries, as well as by plant protection institutes' or extension services' officers. Regarding the estimation of the potential socio-economic impacts of *Xf* for the main crops, a set of parameters/indicators was calculated as a proxy of the consequences of yield loss (Appendix A, Tables A1 and A2) and based on secondary data that were gathered from the European Food Safety Authority's (hereafter EFSA) Scientific Opinion [35] and existing national and international datasets and information sources [36,37].

### 2.2. Assessment Method for Risk of Xf's Potential Establishment and Spread

The risk assessment was performed according to the methodology developed by Cardone et al. [18] and through a specific questionnaire (Supplementary Material S1) involving experts in phytosanitary services and/or quarantine services in each target country. The questionnaire had six questions. Three different options were proposed for each question, corresponding to high, medium, or low risk, with scores ranging from 6 or 3 to 1, respectively [38]. Weights, based on the Delphi technique and established by a focus group of experts [39], were assigned to each question using coefficients in relation to how much each response impacted the overall potential risk, where all weights summed to 1. Therefore, to determine the risk of establishing and spreading the bacterium in each country, the "score × partial coefficient" products of each of the six questions were summed up. Considering that the score for each of the six questions in the questionnaire varied from 1 to 6 and after reweighting, the total scores could range between 2 and 12, we considered that countries with risk values ranging from 2.0 to 4.0, from 4.1 to 6.0, from 6.1 to 12.0 were ranked as low, medium, and high risks of *Xf* invasion, respectively.

### 2.3. Estimation Approach of The Potential Socio-Economic Impacts of Xf
2.3.1. Methodological Considerations

The general scenario assumptions common to all target crops (olives, grapevines, almonds, and *Citrus* spp.) are reported hereafter and reflect those applied by the EFSA Working Group on EU Priority Pests [35] (pp. 128–129) as follows: *"(i) Impacts are assessed by assuming that the entry, establishment and spread of Xf had already occurred; (ii) It is assumed that Xf is not only present throughout the area of potential distribution but also that the circunsumption limits to this area do not change; (iii) Where Xf occurs, it has reached its maximum potential abundance based on current environmental conditions and current crop production practices; (iv) The maximum potential abundance is considered as the driving factor for the estimation of yield/quality loss; (v) Cropping practices and management options are those currently in place in*

*the area of potential distribution and, (vi) Future changes in agricultural practice have not been taken into account".*

### 2.3.2. Yield Losses Estimation

Based on the considerations above, the yield loss indicator was initially estimated in all target crops as expressed by EFSA [35] in terms of yield losses (Table A2). To fit the risk of establishment and spread score values (obtained from the questionnaire—Supplementary Material S1) on the yield loss (%) on main crops reported by the EFSA scientific opinion, percentiles were rescaled into the adopted scale range (1–6). Later, risk scores were rescaled back into yield loss percentiles for each of the main host crops according to Table 1.

**Table 1.** The fitted values of the uncertainty distribution on the yield loss (%) on olive, grapes, *Citrus* spp., and almonds take into consideration the uncertainty range and percentiles of EFSA [35] as shown in Table A1.

| Country | Percentile (in %) | | | | | | | | | | | | | | |
|---|---|---|---|---|---|---|---|---|---|---|---|---|---|---|---|
| | 1 | 2.5 | 5 | 10 | 17 | 25 | 33 | 50 | 67 | 79 | 83 | 90 | 95 | 97.5 | 99 |
| | Fitted Establishment and Spread Score Values (Scale Range: 1–6) | | | | | | | | | | | | | | |
| | 1.05 | 1.13 | 1.25 | 1.50 | 1.85 | 2.25 | 2.65 | 3.50 | 4.35 | 4.75 | 5.15 | 5.50 | 5.75 | 5.88 | 5.95 |
| | Associated percentage loss in yield based on scale range score | | | | | | | | | | | | | | |
| Algeria | | | | 1.70 | | | | | | | | | | | |
| Egypt | | | | | 2.10 | | | | | | | | | | |
| Libya | | | | 1.40 | | | | | | | | | | | |
| Jordan | | | | | | | 1.70 | | | | | | | | |
| Tunisia | | | | | | | 3.00 | | | | | | | | |
| Morocco | | | | | | | | 3.25 | | | | | | | |
| Palestine state | | | | | | | | 3.45 | | | | | | | |
| Lebanon | | | | | | | | 3.75 | | | | | | | |
| Syria | | | | | | | | | 4.35 | | | | | | |
| Albania | | | | | | | | | | 4.80 | | | | | |
| Bosnia and Herzegovina | | | | | | | | | 3.45 | | | | | | |
| Croatia | | | | | | | | | | 4.00 | | | | | |
| France | | | | | | | | | | 4.20 | | | | | |
| Greece | | | | | | | | | | 4.50 | | | | | |
| Israel | | | | | | 2.40 | | | | | | | | | |
| Italy | | | | | | | | | | 4.50 | | | | | |
| Montenegro | | | | | | | | | | 4.50 | | | | | |
| North Macedonia | | | | | | | | | 2.50 | | | | | | |
| Portugal | | | | | | | | | | 4.50 | | | | | |
| Serbia | | | | | 1.70 | | | | | | | | | | |
| Slovenia | | | | | | | | | 3.75 | | | | | | |
| Spain | | | | | | | | | | 4.50 | | | | | |
| Turkey | | | | | | | | | | 4.00 | | | | | |
| Olive trees (<30 years) | 9.4 | 12.1 | 14.9 | 18.5 | 22.0 | 25.6 | 28.7 | 34.6 | 40.9 | 44.5 | 48.9 | 53.6 | 59.0 | 63.5 | 68.5 |
| Olive trees (>30 years) | 24.4 | 30.6 | 36.3 | 43.4 | 49.8 | 55.8 | 60.7 | 69.1 | 76.7 | 80.5 | 84.6 | 88.4 | 91.9 | 94.3 | 96.3 |
| Wine grapes | 0.2 | 0.3 | 0.5 | 0.7 | 0.95 | 1.2 | 1.5 | 2.1 | 2.8 | 3.3 | 3.9 | 4.7 | 5.6 | 6.8 | 8.1 |
| Table grapes | 0.0 | 0.1 | 0.1 | 0.2 | 0.4 | 0.5 | 0.7 | 1.0 | 1.5 | 1.9 | 2.3 | 2.9 | 3.7 | 4.4 | 5.4 |
| *Citrus* spp. | 0.1 | 0.3 | 0.7 | 1.5 | 2.8 | 4.5 | 6.4 | 10.9 | 16.2 | 19.4 | 23.1 | 26.7 | 30.2 | 32.5 | 34.4 |
| Almonds | 1.8 | 2.8 | 3.9 | 5.5 | 7.2 | 8.9 | 10.4 | 13.3 | 16.2 | 17.7 | 19.5 | 21.2 | 22.8 | 24.0 | 25.0 |

Source: Our adaptation based on EFSA [35]. * Risk scores range from 1 (lowest risk) to 6 (highest risk). The top part of this Table was used as a conversion tool between risk scores and percentiles.

### 2.3.3. Economic Assessment Impact

The economic impacts of the losses of important crops in Table A1 can be grouped into: (i) production factors (productivity; value of production; agricultural added value); (ii) marketing factors (trade: import and export; consumption). The parameter "productivity" was proxied by yield and production variation due to the potential outbreak of *Xf*. We used the yield loss coefficient (Table 1) to calculate potential yield loss (tons/ha) and production (tons). All predicted values were based on average values (for a period of 5 years) to minimise productive fluctuation due to biotic and abiotic stresses.

For olives, the groves' age in each country as provided by EFSA [35] was considered when estimating yield loss. The predicted production loss (tons) for olives was estimated by taking into account the share of harvested olive area older and younger than 30 years and based on their average over the period 2015–2019. The lack of data related to the olive groves' age structure led to a straight calculation considering that groves planted after 1989 must be younger than or equal to 30 years. In other words, the harvested area younger than 30 years old was estimated by subtracting from the total olive-growing surface of 2019 that of 1989, which represented the current olive trees older than 30 years. We chose to examine the potential impacts to olive grove distributions in an earlier period (1980–2010), as there was an *Xf* outbreak in Apulia in 2013. Grape production was separated into wine and table grapes based on data from EFSA [35] and the International Organisation of Vine and Wine [40]. Predicted yield (tons/ha) and production losses (tons) for table and wine grapes and *Citrus* spp. were estimated for each country. For *Citrus* spp., the following species and groups were included in the average calculation: citrus such as grapefruit (including pomelos), lemons and limes, oranges, tangerines, mandarins, clementine, and satsumas. All parameters were estimated based on their average over the period 2015–2019.

The *"Value of Production"* parameter was elaborated by considering producer prices and production. The prices were expressed as average (2015–2019) in USD/tons to account for their fluctuation. For some countries in the MENA region, the problem of missed information on prices was solved using the average of the prices of the MENA countries for which producer prices were available. The values of production loss (USD/tons) were calculated by multiplying the quantity of the production loss per price. Furthermore, the economic parameter used as an indicator of profitability were the "Agricultural Value Added" at the farm level corresponding to the gross margin (hereafter GM). The latter is the difference between gross income and variable costs (inputs). Gross income or gross revenue is the total income, cash and noncash, received from an enterprise or business, before any expenses are paid; the variable costs are costs that occur only if the production takes place and that tend to vary with the level of production (direct costs) [41]. These data were obtained from the Sicily region [37], which provides average economic data calculated by Italian Agriculture Accounting Network Italia in the frame of the Farm Accountancy Data Network (hereafter FADN) in Europe. We thus assumed that these Sicilian data were representative of the entire study region. The GM per hectare was calculated for each target crop in pre-*Xf* and post-*Xf* situations (entry, establishment, and spread) in the target countries. The reduction of revenues and variable costs per hectare and year was calculated taking into consideration the needs of inputs (i.e., fertilisers, pesticides, fuel, etc.) for the most important and relevant operations in the field, such as pruning, fertilisation, pest management, irrigation, and harvesting, according to the estimated loss of production. The above GM per hectare and the production loss pre-*Xf* and post-*Xf* were used to calculate the loss of GM in percentage. The loss of GM, corresponding to the relative loss of production, was recalculated based on the yield loss in each target country for the assessment of the economic impact of *Xf* at the farm level.

The potential impact on *"Trade"* was based on the effect of production reduction on the quantity of the crops exported by each country. The average export quantity based on the average production was reduced according to the reduction (%) of production due to the presence of *Xf*. The potential impact on trade did not take into account bans or trade restrictions but was considered as a direct consequence of the decreased production.

The *"Consumption"* parameter was based on export, import, stock variation, and production data that were retrieved from the FAOSTAT database [36]. The absence of data on supply elasticity forced us to use a direct approach, considering the effect on consumption (supply) by keeping the average value of import quantity and stock variation constant and considering the decrease in production and exports due to *Xf*. Higher imports are quantities that have to be imported to keep consumption constant at the value before the *Xf* outbreak, where no substitution effects apply; it was calculated by increasing the pre-*Xf* initial value equal to the percentage of production loss.

#### 2.3.4. Social Impact Assessment

The social assessment impact was based on two parameters/indicators: (i) loss of employment and (ii) social vulnerability index. The loss of production induced a work reduction expressed in terms of hours per hectare and year and was estimated by taking into account the labour requirements for the most important and relevant operations in the field, such as pruning and harvesting. Italian references [42,43] and FADN database [44] were used by experts in the calculation. Regarding the olive sector, work loss was calculated in two different types of olive groves: the first was an intensive grove with olive trees less than 30 years old, with trees spaced $5 \times 5$ m or $6 \times 6$ m apart, while the second was an extensive olive grove with olive trees over 30 years old, with trees spaced $7 \times 7$ m or $8 \times 8$ m apart. The overall loss of employment in hours and days per crop and per target country was assessed by multiplying the loss of employment per hectare by the total area harvested of each target crop. Here, it was considered approximately that one day was equal to seven hours depending on national agreement between entrepreneurs and workers. In addition, we explored the assessment of *social vulnerability*, in particular the impacts on small farms. Social vulnerability can be linked to the migration from small rural communities to urban areas or out of the country, following changes in agriculture yield and profitability. As such, a new complex "Social vulnerability index" was created which considers four components (indices): (i) employment in agriculture [45] that considers the weight of agricultural jobs on the total employment; (ii) the Global Food Security index [46] that considers the issues of food affordability, availability, quality and safety, and natural resources and resilience; (iii) the average size per agricultural holding [47] that takes into account the presence of large, medium, and small farms, and the weaknesses of marginal communities composed mainly of small farmers; and (iv) the gross national income (GNI) per capita [48] that is the country's economic strengths and needs, as well as the general standard of living enjoyed by the average citizen. GNI tends to be closely linked with other indicators that measure the social, economic, and environmental well-being of the country and its people. Meanwhile, the Global Food Security Index (hereafter GFI) was divided into its four subcomponents: GFI—affordability; (ii) GFI—availability; (iii) GFI—quality and safety; and (iv) GFI—natural resources and resilience. The experts assigned the weights (%) to the indices above to calculate the new index for each country regarding the greater or lower influence on social vulnerability. In the end, in order to assess the risk of *Xf*'s establishment and spread in the target countries, the values of the above indices were multiplied by the pest risk management values (Table 1) estimated for each country in this study.

### 3. Results

#### 3.1. Risk of Xf's Potential Establishment and Spread

Figure 1 highlights the ranking score of the Balkans, European Mediterranean, and MENA countries according to six establishments and spread risk indicators. It shows the position of each country involved compared to the others in the risk spectrum. Libya (1.4/6) appears to be the least vulnerable country, while Albania (4.8/6) is ranked at the highest risk level for potential exposure to an invasion by *Xf*. In Albania, the regulatory status of *Xf* and the relative abundance of alternative hosts may explain the highest level of exposure to the potential *Xf* establishment and spread. In Libya, the unsuitable climate conditions, the low presence of *Xf* vectors, the level of regulatory status against *Xf*, and the relatively low availability of hosts explain this low level of risk. Regarding the European Balkan region, Croatia (4/6) and Montenegro (4/6) are ranked at the high potential establishment and spread risk level, followed by Bosnia and Herzegovina (3.45/6) and Slovenia (3.25/6). Serbia (1.7/6) appears as the least exposed country to the spread of *Xf* due mainly to the low abundance level of the main vulnerable crops in agricultural land. In the European Mediterranean region, most of the countries (4.5/6) involved in the study are at the high to highest risk levels, except France (4.2/6), due to the level of relative abundance of major crops in agricultural land (0.3/6 in France, and 0.6/6 in Greece, Italy and Portugal).

As regards the MENA region, Syria (4.35/6) and Turkey (4/6) are at the highest level of risk, followed by Lebanon (3.7/6), Palestine (3.45/6), Morocco (3.25/6), Tunisia (3/6), Israel (2.4/6), Egypt (2.1/6), Jordan (1.7/6), Algeria (1.7/6), and Libya (1.4/6). Due to the internal political instability, Syria could not organize a certification program for plant propagation material and is therefore considered at high risk of *Xf* establishment and spread in the MENA region. All the other countries in this region have a more or less advanced certification system and are therefore more prepared to counter the spread of this plant disease.

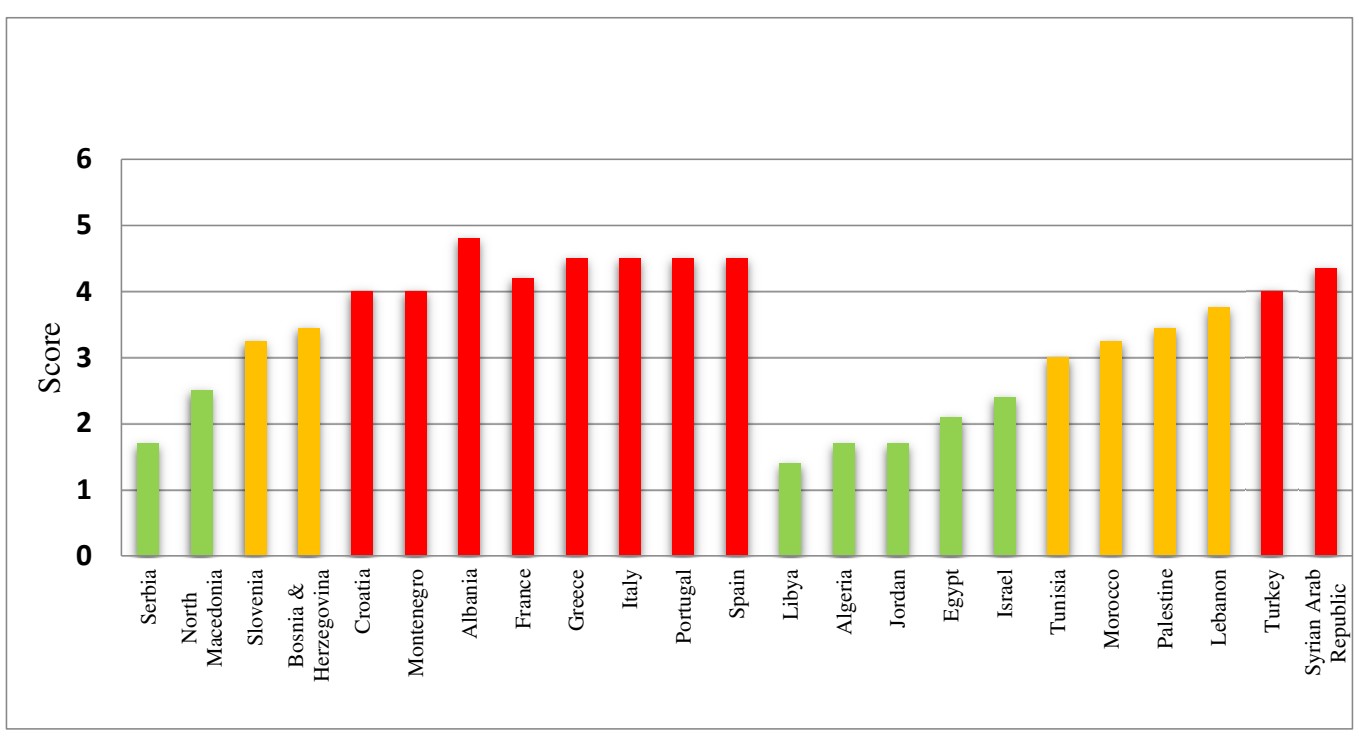

**Figure 1.** Risk scores of the European Balkans, European Mediterranean, and MENA countries according to six establishments and spread risk indicators. Green, yellow and red colours refer to low, medium, and high risk scores, respectively.

### 3.2. Potential Economic Impacts of Xf on Target Crops

In the European Mediterranean countries, the total quantity of production loss (Table 2) was estimated at around 11.06 million tons (of which 78% for olives, 5.3% for grapes, 16.1% for *Citrus* spp., and 0.6% for almonds). It represents 24.45% of the harvested production (e.g., olives, grapes, *Citrus* spp., and almonds). The total value of production loss was estimated at USD 12.44 billion (of which 75.7% for olives, 5.7% for grapes, 17.7% for *Citrus* spp., and 0.9% for almonds). It represents 23.21% of the total value of the harvested production. The overall loss of gross margin (Table 3) at the farm level was more than USD 8.4 billion (of which 75.6% for olives, 8.6% for grapes, 9.9% for *Citrus* spp., and 5.8% for almonds). It represents 43.2% of the total gross margin of the harvested production. The total export loss was estimated at around 3.98 million tons (of which 0.3% for olives, 15.3% for grapes, 84.2% for *Citrus* spp., and 0.2% for almonds). It represents 83.25% of the total exports of the production harvested. The total quantity of imports increase was rated at around 31.65 million tons (of which 83.7% for olives, 5.5% for grapes, 10% for *Citrus* spp., and 0.9% for almonds). It represents 32% of the total imports of the production harvested.

**Table 2.** Economic impacts of *Xylella fastidiosa* on olives, grapes, *Citrus* spp. and almonds in the Mediterranean basin.

| Region | Country | Production Loss (Average Values of 2015–2019) | |
|---|---|---|---|
| | | In Tons | In USD |
| European Balkans | Albania | 93,584 | 134,054,631 |
| | Bosnia and Herzegovina | 633 | 337,761 |
| | Croatia | 22,448 | 28,643,729 |
| | Montenegro | 2625 | 0 |
| | North Macedonia | 6382 | 759,827 |
| | Serbia | 889 | 554,609 |
| | Slovenia | 2643 | 1.194,567 |
| **Subtotal** | | **129,204** | **165,545,124** |
| European Mediterranean | France | 189,886 | 434,440,258 |
| | Greece | 1,685,631 | 3,938,480,439 |
| | Italy | 2,347,814 | 2,273,802,603 |
| | Portugal | 655,615 | 405,854,601 |
| | Spain | 6,184,653 | 5,392,861,391 |
| **Subtotal** | | **11,063,653** | **12,445,439,319** |
| MENA | Algeria | 290,334 | 407,274,281 |
| | Egypt | 492,376 | 174,969,969 |
| | Jordan | 67,290 | 87,553,802 |
| | Lebanon | 122,114 | 207,477,909 |
| | Libya | 41,372 | 41,393,923 |
| | Morocco | 917,538 | 585,755,556 |
| | Palestine | 34,411 | 68,192,398 |
| | Syria | 656,283 | 827,460,116 |
| | Tunisia | 547,732 | 308,736,982 |
| | Israel | 48,830 | 99,271,118 |
| | Turkey | 1,845,693 | 1,727,804,586 |
| **Subtotal** | | **5,063,993** | **4,535,890,640** |

**Table 3.** Economic impacts of *Xylella fastidiosa* in terms of gross margin on olives, grapes, *Citrus* spp., and almonds in the Mediterranean basin.

| Region | Country | Gross Margin Loss (Average Values of 2015–2019) |
|---|---|---|
| European Balkans | Albania | 37,028,027 |
| | Bosnia and Herzegovina | 1,694,147 |
| | Croatia | 19,826,607 |
| | Montenegro | 1,632,292 |
| | North Macedonia | 6,497,631 |
| | Serbia | 4,972,013 |
| | Slovenia | 5,405,747 |
| **Subtotal** | | **77,057,094** |
| European Mediterranean | France | 260,426,626 |
| | Greece | 1,097,022,789 |
| | Italy | 1,941,725,143 |
| | Portugal | 547,338,626 |
| | Spain | 4,538,900,744 |
| **Subtotal** | | **8,385,413,928** |
| MENA | Algeria | 184,646,860 |
| | Egypt | 64,845,989 |
| | Jordan | 33,294,057 |
| | Lebanon | 73,572,082 |
| | Libya | 119,212,449 |
| | Morocco | 754,985,614 |
| | Palestine | 0 |
| | Syria | 0 |
| | Tunisia | 646,052,940 |
| | Israel | 53,644,139 |
| | Turkey | 706,513,255 |
| **Subtotal** | | **2,636,767,384** |

Regarding the Balkans, the total quantity of production loss was estimated at 0.13 million tons (of which 75.6% for olives, 11.6% for grapes, 12.7% for *Citrus* spp., and 0.1% for almonds). It represents 10.99% of the total the production harvested (olives, grapes, *Citrus* spp., and almonds). The total value of production loss was estimated at USD 165.54 million (of which 78.3% for olives, 5% for grapes, 16.6% for *Citrus* spp., and 0.1% for almonds). It represents 20.09% of the total value of the production harvested (olives, grapes, *Citrus* spp., and almonds). The overall loss of gross margin at the farm level was about USD 77.0 million (of which 62.4% for olives, 30.5% for grapes, 6.5% for *Citrus* spp., and 0.6% for almonds). It represents 14.1% of the total gross margin of the production harvested. The total quantity of export loss was estimated at around 0.07 million tons (of which 0.1% for olives, 62.9% for grapes, 37% for Citrus spp., and 0.1% for almonds). It represents 63.34% of the total exports of the harvested production. The increase in the total quantity of imports was rated at around 0.11 million tons (of which 78.6% for olives, 12% for grapes, 9.3% for Citrus spp., and 0.1% for almonds). It represents 12% of the total value of imports of the production harvested (Tables 2 and 3).

Concerning the Middle East and North Africa region, the total quantity was estimated at around 5.06 million tons (of which 67.9% for olives, 2% for grapes, 28.7% for *Citrus* spp., and 1.2% for almonds). It represents 16.98% of the total production harvested (olives, grapes, *Citrus* spp., almonds). The total value of production loss was estimated at around USD 4.53 billion (of which 63.3% for olives, 1.23% for grapes, 33.9% for *Citrus* spp., and 1.5% for almonds). It represents 16.1% of the total value of the production harvested. The overall loss of gross margin at the farm level was about $2.6 billion (of which 77.0% for olives, 5.3% for grapes, 11.8% for *Citrus* spp., and 5.9% for almonds).

It represents 17.0% of the total GM of the production harvested. The total export loss was estimated at around 3.83 million tons (of which 0.6% for olives, 8.8% for grapes, 90.5% for Citrus spp., and 0.1% for almonds). It represents 90.2% of the total exports of the production harvested. The increase in the total quantity of imports was rated at around 3.56 million tons (of which 65.3% for olives, 2.8% for grapes, 29.6% for *Citrus* spp., and 2.2% for almonds). It represents 18% of the total value of imports of the production harvested.

Overall, the economic impact of the loss of production value (gross production value—USD) of all target crops on total agricultural production (excluding the value of livestock) in MED countries was assessed. Table 4 shows that the highest percentage, equal to 10.4%, was in the European Mediterranean area; in particular, it was the highest in Greece 34.0% and in Spain (17.1%). The percentage was 4.4% in the MENA area, the highest in Morocco (10.0%), followed by Lebanon (9.4%) and Palestine (8.7%), and 2.1% in the European Balkans, with the highest percentage in Albania (9.8%).

### 3.3. Potential Social Impacts of Xf on Target Crops

In the European Mediterranean countries, the total amount of employment loss was appraised at 203.93 million days (of which 86.6% for olives, 28.3% for grapes, 3.3% for Citrus spp., and 1.8% for almonds). In the Balkans area, the total amount of employment loss was appraised at 2.68 million days (of which 79.1% for olives, 18.2% for grapes, 2.4% for Citrus spp., and 0.2% for almonds). Concerning the Middle East and North Africa region, the total amount of employment loss was appraised at 131.2 million days (of which 91.1% for olives, 2.3% for grapes, 4.4% for *Citrus* spp. and 2.1% for almonds).

In terms of the social vulnerability assessment on small farms, Tables 5 and 6 reveal that in 2019, the values of the social vulnerability index in MENA countries were higher than in European MED countries. Currently, it is 4.3 in Morocco, 4.2 in Syria, and 4.0 in Turkey in the absence of *Xf*. Considering the effects of the risk of *Xf*'s establishment and spread in MENA countries, the post-*Xf* social vulnerability changed significantly. In fact, it reached a very high value in Syria, Turkey, Morocco, and Tunisia. It could mean that with all the precautions that the use of the index imposes, the establishment and spread of *Xf* in those countries could significantly impact the social conditions especially for small farms. If we consider the effects of the risk of establishment and spread of *Xf* in the EU

MED countries on the social vulnerability, it could also be noticed that in the EU MED countries, the social vulnerability index changed significantly, especially in Greece, and that the differences between the two regions were reduced. However, we stress that this last result was essentially due to the high value of the pest risk index in the EU MED countries, much higher than in some MENA countries (1.7 in Algeria vs. 4.5 in Italy).

**Table 4.** Loss of production of *Xylella fastidiosa* on olives, grapes, *Citrus* spp., and almonds in the Mediterranean basin.

| Region | Country | Loss of Production in % (Average Values of 2015–2019) | |
| --- | --- | --- | --- |
| | | Agriculture Value (in USD, Livestock Excluded) | % |
| European Balkans | Albania | 1,361,171,800 | 9.85 |
| | Bosnia and Herzegovina | 1,156,920,400 | 0.03 |
| | Croatia | 1,037,008,600 | 2.76 |
| | Montenegro | - | - |
| | North Macedonia | 791,495,800 | 0.10 |
| | Serbia | 3,340,724,000 | 0.02 |
| | Slovenia | 361,000,000 | 0.33 |
| Subtotal | | 8,048,320,600 | 2.06 |
| European Mediterranean | France | 42,035,106,000 | 1.03 |
| | Greece | 11,645,832,000 | 33.82 |
| | Italy | 30,309,234,000 | 7.50 |
| | Portugal | 4,212,530,600 | 9.63 |
| | Spain | 31,519,855,200 | 17.11 |
| Subtotal | | 119,722,558,200 | 10.40 |
| MENA | Algeria | 16,645,583,600 | 2.45 |
| | Egypt | 17,512,204,600 | 1.00 |
| | Jordan | 1,313,911,200 | 6.66 |
| | Lebanon | 2,216,981,200 | 9.36 |
| | Libya | - | - |
| | Morocco | 8,588,718,000 | 6.82 |
| | Palestine | 785,026,400 | 8.69 |
| | Syria | - | - |
| | Tunisia | 3,091,055,600 | 9.99 |
| | Israel | 3,933,751,600 | 2.49 |
| | Turkey | 50,437,915,800 | 3.43 |
| Subtotal | | 104,585,147,400 | 4.34 |

*3.4. Potential Social Impacts of Xf on Target Crops*

In the European Mediterranean countries, the total amount of employment loss was appraised at 203.93 million days (of which 86.6% for olives, 28.3% for grapes, 3.3% for Citrus spp., and 1.8% for almonds). In the Balkans area, the total amount of employment loss was appraised at 2.68 million days (of which 79.1% for olives, 18.2% for grapes, 2.4% for Citrus spp., and 0.2% for almonds). Concerning the Middle East and North Africa region, the total amount of employment loss was appraised at 131.2 million days (of which 91.1% for olives, 2.3% for grapes, 4.4% for *Citrus* spp. and 2.1% for almonds).

In terms of the social vulnerability assessment on small farms, Tables 5 and 6 reveal that in 2019, the values of the social vulnerability index in MENA countries were higher than in European MED countries. Currently, it is 4.3 in Morocco, 4.2 in Syria, and 4.0 in Turkey in the absence of *Xf*. Considering the effects of the risk of *Xf*'s establishment and spread in MENA countries, the post-*Xf* social vulnerability changed significantly. In fact, it reached a very high value in Syria, Turkey, Morocco, and Tunisia. It could mean that with all the precautions that the use of the index imposes, the establishment and spread of *Xf* in those countries could significantly impact the social conditions especially for small farms. If we consider the effects of the risk of establishment and spread of *Xf* in the EU MED countries on the social vulnerability, it could also be noticed that in the EU MED countries, the social vulnerability index changed significantly, especially in Greece, and that the differences between the two regions were reduced. However, we stress that this last

result was essentially due to the high value of the pest risk index in the EU MED countries, much higher than in some MENA countries (1.7 in Algeria vs. 4.5 in Italy).

**Table 5.** Assessment of the social vulnerability index due to *Xylella fastidiosa* on olives, grapes, *Citrus* spp., and almonds in the Middle East and North Africa countries (Year: 2019).

| Index | Partial Index to Social Vulnerability Index | Weight | Palestine Risk score | Palestine Risk weighted | Syria Risk score | Syria Risk weighted | Egypt Risk score | Egypt Risk weighted | Libya Risk score | Libya Risk weighted | Tunisia Risk score | Tunisia Risk weighted | Jordan Risk score | Jordan Risk weighted | Algeria Risk score | Algeria Risk weighted | Lebanon Risk score | Lebanon Risk weighted | Morocco Risk score | Morocco Risk weighted | Turkey Risk score | Turkey Risk weighted | Israel Risk score | Israel Risk weighted |
|---|---|---|---|---|---|---|---|---|---|---|---|---|---|---|---|---|---|---|---|---|---|---|---|---|
| 1 | Agricultural employment | 0.20 | 6 | 1.2 | 3 | 0.6 | 6 | 1.2 | 5 | 1 | 4 | 0.8 | 2 | 0.4 | 3 | 0.6 | 4 | 0.8 | 6 | 1.2 | 5 | 1.0 | 1 | 0.2 |
| 2 | GFI affordability | 0.10 | | 0.0 | 4 | 0.4 | 3 | 0.3 | | 0.0 | 2 | 0.2 | 1 | 0.1 | 1 | 0.1 | | 0.0 | 1 | 0.1 | 2 | 0.2 | 1 | 0.1 |
| | GFI availability | 0.10 | | 0.0 | 4 | 0.4 | 1 | 0.1 | | 0.0 | 3 | 0.3 | 3 | 0.3 | 3 | 0.3 | | 0.0 | 3 | 0.3 | 2 | 0.2 | 1 | 0.1 |
| | GFI quality and safety | 0.05 | | 0.0 | 3 | 0.2 | 2 | 0.1 | | 0.0 | 2 | 0.1 | 2 | 0.1 | 2 | 0.1 | | 0.0 | 2 | 0.1 | 1 | 0.1 | 1 | 0.1 |
| | GFI—natural resources and resilience | 0.05 | | 0.0 | 4 | 0.2 | 3 | 0.2 | | 0.0 | 3 | 0.2 | 3 | 0.2 | 4 | 0.2 | | 0.0 | 3 | 0.2 | 3 | 0.2 | 3 | 0.2 |
| 3 | Average size per agr. holding | 0.20 | 5 | 1.0 | 3 | 0.6 | 1 | 0.2 | 1 | 0.2 | 1 | 0.2 | 1 | 0.2 | 1 | 0.2 | 1 | 0.2 | 3 | 0.6 | 3 | 0.6 | 1 | 0.2 |
| 4 | Gross national income per capita | 0.30 | 6 | 1.8 | 6 | 1.8 | 6 | 1.8 | 6 | 1.8 | 6 | 1.8 | 6 | 1.8 | 6 | 1.8 | 6 | 1.8 | 6 | 1.8 | 6 | 1.8 | 6 | 1.8 |
| | Total social vulnerability index | 1 | n.a. | | 4.2 | | 3.9 | | n.a. | | 3.6 | | 3.1 | | 3.3 | | n.a. | | 4.3 | | 4.0 | | 2.6 | |
| 5 | Pest risk (establishment and spread) | - | 3.5 | | 4.4 | | 2.1 | | 1.4 | | 3.0 | | 1.7 | | 1.7 | | 3.8 | | 3.3 | | 4.0 | | 2.4 | |
| 6 | Social vulnerability index post-*Xf* | - | n.a. | | 18.1 | | 8.1 | | n.a. | | 10.7 | | 5.2 | | 5.6 | | n.a. | | 13.8 | | 16.0 | | 6.2 | |

**Table 6.** Assessment of the social vulnerability index due to *Xylella fastidiosa* on olives, grapes, *Citrus* spp. and almonds in the European Mediterranean countries (Year: 2019).

| Index | Partial Index to Social Vulnerability Index | Weight | France Risk score | France Risk weighted | Greece Risk score | Greece Risk weighted | Italy Risk score | Italy Risk weighted | Portugal Risk score | Portugal Risk weighted | Spain Risk score | Spain Risk weighted |
|---|---|---|---|---|---|---|---|---|---|---|---|---|
| 1 | Agricultural employment | 0.2 | 2 | 0.4 | 4 | 0.8 | 2 | 0.4 | 3 | 0.6 | 2 | 0.4 |
| 2 | GFI affordability | 0.1 | 1 | 0.1 | 1 | 0.1 | 1 | 0.1 | 1 | 0.1 | 1 | 0.1 |
| | GFI availability | 0.1 | 2 | 0.2 | 2 | 0.2 | 1 | 0.1 | 2 | 0.2 | 2 | 0.2 |
| | GFI quality and safety | 0.05 | 1 | 0.05 | 1 | 0.05 | 1 | 0.05 | 1 | 0.05 | 1 | 0.05 |
| | GFI—natural resources and resilience | 0.05 | 2 | 0.1 | 3 | 0.15 | 3 | 0.15 | 3 | 0.15 | 2 | 0.1 |
| 3 | Average size per agr. holding | 0.2 | 1 | 0.2 | 4 | 0.8 | 2 | 0.4 | 1 | 0.2 | 1 | 0.2 |
| 4 | Gross national income per capita | 0.3 | 1 | 0.3 | 1 | 0.3 | 1 | 0.3 | 1 | 0.3 | 1 | 0.3 |
| | Total social vulnerability index | 1 | 1.35 | | 2.4 | | 1.5 | | 1.6 | | 1.35 | |
| 5 | Pest risk (establishment and spread) | | 4.2 | | 4.5 | | 4.5 | | 4.5 | | 4.5 | |
| 6 | Social vulnerability index post-*Xf* | | 5.67 | | 10.80 | | 6.75 | | 7.20 | | 6.08 | |

## 4. Discussion

The findings explored in this study provided a panoramic picture of the risk of establishment and spread of the bacterium and its potential socio-economic impacts in the European Mediterranean (France, Greece, Italy, Portugal, and Spain), the Balkans (Albania, Montenegro, Croatia, Serbia, Bosnia-Herzegovina, and North Macedonia), and the Middle East and North African (Algeria, Egypt, Israel, Jordan, Lebanon, Libya, Morocco, Palestine, Syria, Tunisia, and Turkey) countries. In this paper, the difference in the risk of the bacterium establishment and spread in these areas appeared heterogeneous due to several risk drivers. In this perspective, climate suitability is likely to be an important risk driver considering that the *Xf* growth rate is known to be sensitive to temperature. In fact, the survival of *Xf* in grapevines was affected below 12 to 17 °C and above 34 °C, whereas its growth rate was rapid between 25 and 32 °C, with an optimum of 28 °C, for the epidemiology of Pierce's disease [49]. In this context, previous studies examined this risk driver. They revealed that most countries from the Mediterranean basin were highly suitable. At the same time, those in the north of the EU and Gulf Arab region were highly unsuitable for the potential establishment and spread of *Xf* according to quarterly summer temperatures. In the EU and the MENA region, the most vulnerable country with respect to climate suitability was Malta, followed by Lebanon, Greece, Portugal, Algeria, Spain, Turkey, Egypt, Morocco, and Albania. The North European and Arabian Gulf countries were thus less vulnerable to the spread of *Xf*. Further risk assessment studies [13,14,17,49–52] of this bacterium, mainly based on climate suitability indicators, showed a high suitability in the demarcated zones in Europe except for Germany, where *Xf* did not spread as well as in the Mediterranean basin.

Furthermore, it is of utmost importance for each country to deal with the problem immediately and implement preventive measures with appropriate means, aiming to control this extremely dangerous pathogen. Regarding the assessment of the socio-economic risk, it is necessary to highlight that this paper is one of the first simultaneous explorations of the potential economic impact of *Xf* over all these regions. A range of quantitative economic impact assessment methods (partial budgeting method, partial equilibrium modelling, input–output analysis, general equilibrium modelling) can be used in pest risk analysis to forecast the direct and indirect market impacts of an alien species invasion such as *Xf* in a new location. The required quantity of data and expertise increase sharply from the first to the last technique listed. However, it is beyond the scope and the geographic scale of this study to consider even one of these three techniques. Partial budgeting principles were used to estimate the production parameters. The literature regarding the effects on trade and consumption of the pest invasion uses partial or general equilibrium models, taking into consideration the elasticity and price variation, as well as the impact on other nonagricultural markets or macroeconomic changes [1,34]. An input–output analysis explores the interdependence between different sectors of a single national economy or different province economies. All this goes beyond the scope of this study which, instead, intended to provide one of the first insights into the socio-economic impacts of *Xf* in some target countries. The socio-economic impact assessment resulted in a survey exploration of the area at risk, with specific and strong outcomes in terms of declining yields, production, profitability, trade (see export), employment, increasing imports to keep the same levels of consumption, and social vulnerability.

The absence of such a study is a critical constraint to let policymakers mitigate its potential severe impacts. Recent studies estimated the overall economic costs of invasive species at USD 25.3 billion and USD 139.5 billion in 2020 in the Mediterranean basin [53] and Europe [54], respectively. Regarding the studied bacterium, previous papers estimated the impacts of *Xf* outbreaks. In Italy, the infected area by *Xf* expanded from about 8000 ha in 2013 to 715,000 ha in 2018. The borders of the areas currently declared "infected" cover almost 36% of the Apulia region (Southern Italy), with about 21 million olive trees under the threat of bacterial infection [6]. Nevertheless, the olive-growing area was completely lost due to the *Xf* presence, representing 14.06% of the Apulian olive-growing area and

4.61% of the national total. Apulia's production loss in a three-year period (2016–2018) was estimated at 29,000 tons (equivalent to EUR 390 million), representing 10% of the Italian olive production [55]. The mean value of loss of the socio-ecological benefits was 1059 EUR/ha [20]. The bacterium had also severe impacts on other countries and regions. In Brazil, the wide distribution and high prevalence of the vectors favoured *Xf*'s dissemination, which caused disease in 25% of citrus trees in 1996 and reached 44% in 2003, when losses of more than USD 100 million per year in citrus production were estimated [56]. In the USA, Pierce's disease in grapevines costs California USD 104 million per year [32]. A recent report by the Joint Research Center (JRC) estimated that due to production losses, the possible spread of *Xf* could eventually cost the EU EUR 5.5 billion per year, with a potential export loss of EUR 700 million per year for olive, citrus, almond, and grape crops in southern and central Italy, France, Spain, and Portugal. In Australia, costs could range from USD 2.3 billion to USD 7.9 billion over 50 years on Australian grapes and wineries [33]. Schneider et al. [57] stressed the importance of strengthening research to reduce the economic impact, ranging from EUR 0.6 billion to EUR 1.6 billion, by replanting resistant cultivars and applying phytosanitary measures; moreover, the authors reported that the loss of Italian olive production could increase from EUR 1.9 billion to EUR 5.2 billion over 50 years in the worst-case scenario. In the Near East and North Africa region, the values of production losses were estimated at USD 10.0 million, USD 218.35 million, and USD 1.0 billion for grapes, Citrus *spp.*, and olives, respectively [18]. The bacterium also affects ornamental plants such as oleander, which is largely present along the main roads and in private gardens in the USA; losses along Californian highways alone are estimated at USD 125 million [58]. In New Jersey, bacterial leaf scorch was estimated to affect 35% of oaks, causing both aesthetic and economic damages [29].

From this study, policymakers in countries not yet infected by *Xf* should draw inspiration and take further preventive management measures, enhance their legislative regulations, laboratory and equipment resources, networking, communication and awareness campaigns, and conduct continuous surveillance and monitoring to prevent the epidemics of this biological invader species. The present study could be extended to assess the private costs management of a potential *Xf* invasion in the Mediterranean basin, mainly costs related to replantation or business interruption and to analyse the potential compensation measures to be applied to counter the invasion of *Xf* in noninfected areas.

## 5. Conclusions

The present paper constituted a first exploration of the potential impacts based on the risk level for the establishment and spread of *Xf* in Euro-Mediterranean countries, the Balkans and the MENA region, on *Olea europaea* (i.e., olives), *Vitis vinifera* (i.e., grapes), *Citrus* spp., and *Prunus dulcis* (i.e., almonds). The results explored in this study were only a preliminary "explorative" assessment of the socio-economic impacts that gave an idea of the potential scenario of the establishment and spread of *Xf*. The results of this exploratory analysis—which considered the worst-case, yet plausible, scenario in terms of the establishment and spread of *Xf* and did not consider management and compensation measures—strongly suggested that the socio-economic impacts expected from the spread of *Xf*, in this wide scenario, were unacceptable. Further, this study identified specific countries in the MENA region at high risk due to a high social vulnerability.

**Supplementary Materials:** The following supporting information can be downloaded at: https://www.mdpi.com/article/10.3390/d14110975/s1, Supplementary Material S1: The questionnaire for the assessment of the risk of *Xylella fastidiosa* potential establishment and spread on the main crops in the European Mediterranean countries, in the Balkans and in the Middle East North Africa region.

**Author Contributions:** Conceptualization, G.C, M.D., K.D., M.F., C.R. and V.F; methodology, G.C, M.D., K.D., M.F., C.R. and V.F; software, M.F. and C.R.; validation, G.C., M.D., K.D., M.F., C.R. and V.F.; formal analysis, G.C., M.D., K.D., M.F., C.R. and V.F.; investigation, G.C., M.D., K.D., M.F., C.R. and V.F. resources, G.C. and A.L.; data curation, M.F. and C.R.; writing—original draft preparation,

M.F.; writing—review and editing, G.C., M.D., K.D., M.F., C.R. and V.F.; visualization, G.C., M.F. and A.L.; supervision, G.C. and V.F.; project administration, G.C. and A.L.; funding acquisition, G.C. and A.L. All authors have read and agreed to the published version of the manuscript.

**Funding:** This research was funded by SLM Partners Capital LLP (UK), a consultancy organization in the sectors of agriculture, environment, and rural development.

**Institutional Review Board Statement:** Not applicable.

**Data Availability Statement:** Not applicable.

**Acknowledgments:** Thanks to Wanda Occhialini and Elvira Lapedota for the language review. Special thanks go to the Quadature Climate Foundation and the Cibus Foundation for their grant to SLM Partners, which made this study possible.

**Conflicts of Interest:** The authors declare no conflict of interest. The funders had no role in the design of the study; in the collection, analyses, or interpretation of data; in the writing of the manuscript; or in the decision to publish the results.

## Appendix A

Two Tables A1 and A2 are added here as an appendix.

**Table A1.** Estimated yield losses and uncertainty ranges used for the estimation of potential socio-economic impacts of *Xylella fastidiosa* for the main crops in the European Mediterranean countries, the Balkans, and the Middle East and North Africa region.

| Crop (as Stated by EFSA 2019) | Estimated Yield Loss (Median) | 90% Uncertainty Range | |
|---|---|---|---|
| | | 5th Percentile | 95th Percentile |
| Olive trees younger than 30 years | 34.6% | 14.9% | 59.0% |
| Olive trees older than 30 years | 69.1% | 36.3% | 91.9% |
| Wine grape in southern EU | 2.1% | 0.5% | 5.6% |
| Table grape in southern EU | 1.0% | 0.1% | 3.7% |
| *Citrus* spp. | 10.9% | 0.7% | 30.2% |
| Almonds | 13.3% | 3.9% | 22.8% |

Source: EFSA [35].

**Table A2.** Parameters used for the estimation of potential socio-economic impacts of *Xylella fastidiosa* for the main crops in the European Mediterranean countries, the Balkans, and the Middle East and North Africa region.

| Type of Parameter | Indicator Available | Unit | Source | Available Year * |
|---|---|---|---|---|
| | Area harvested | Ha | FAOSTAT | To 2019 |
| Productivity | Yield | Hg/Ha | FAOSTAT | To 2019 |
| | Production | Tons | FAOSTAT | To 2019 |
| Value of production | Gross production value | USD | FAOSTAT | To 2019 |
| | Producer prices | USD/tons | FAOSTAT RDP ** | To 2019 2014–2020 |
| Agricultural value-added | Gross margin | EUR/Ha | FADN | 2014–2020 |
| Employment | Agricultural employment | Hours/Ha | FADN | To 2018 |
| Trade | Import | Tons | FAOSTAT | To 2019 |
| | Export | Tons | FAOSTAT | To 2019 |
| Consumption | Production import Stock variation export | Tons | FAOSTAT | To 2019 |

Source: Our elaboration based on FAOSTAT data [36]. * Data availability changes from one crop to another. ** Rural Development Programme [37].

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
