# Peer review of "Socio-Economic Risks Posed by a New Plant Disease in the Mediterranean Basin"

_diversity, doi:10.3390/d14110975_

Round 1

Reviewer 1 Report

review_diversity

General comments

This manuscript provides a risk analysis at an impressive geographic scale for a pest of high concern. I appreciate the many facets of risk examined and the development of a social vulnerability index. However, the methods lacked a large amount of detail that made it difficult to evaluate the rigour of the work. The text felt quite long and redundant in sections, and the tenses used made it difficult to follow which components were methods, and which were previous findings. I suggest switching from passive to active voice throughout the writing to reduce the wordiness of sentences and make methods more clear, and presenting all methods/results in the form ‘we did/we found’. In the spirit of open science, I also would like to see the equations used to calculate scores and losses in text (in particular the spatial analysis of age-structured groves, as I did not follow this part at all), but ideally the code and data uploaded to GitHub+Zenodo/Figshare or similar. See my detailed comments below. One very important concern I have without having a better understanding of the expert elicitation protocol is that ethics approval was required but not obtained. Please provide more detail on the expert elicitation ethics considerations and refer to the elicitation literature in general (i.e. Delphi protocol) in this section.

The title is grammatically incorrect (should be “invasive” not “invasion”). Suggest ‘Socio-ecomonic risks posed by a new plant disease in the Mediterranean basin’ of similar to appeal to a broader audience.

Abstract

line 22 - please add more information on the nature of the ad hoc approach.

line 26 - ‘the Euro-Mediterranean region’

line 33 - remove ’today’

line 36 - ‘induce’

line 38 - Should this read ‘declared established’?

line 40 - ‘plants for planting’ sounds redundant and unspecific. Do you mean ‘horticultural plants and nursery stock’? If so, reword throughout

line 41 - new paragraph here with a topic sentence identifying that this paragraph discusses pathways of transport

line 41 - ‘African’

lines 43 -45 - this is repetitive with the beginning of this paragraph. Please increase conciseness of the introduction.

line 46 - ‘across geographic barriers’

lines 46 -49 - this again is very repetitive with what’s already been stated

line 54 - country lists should be written here and the MENA acronym should be defined at first mention for the Balkans and MENA rather than below.

line 55 - start another new paragraph here

line 59 - remove italics on ‘was’ and change to ‘has been’

lines 66-68 -change to something along the lines of ‘Due to the high spread potential and high risk of economic and ecological impacts posed by Xf, there is a need..’

line 72 - does ‘this research’ refer to this manuscript? if so, I would rephrase as ‘In this study, we ..’

line 72 - ‘and estimate associated losses’

line 75 - and is present twice in this sentence, should the and on the previous line be converted to a comma and this clause reduced to ‘’and the MENA region”?

lines 78-79 - these lines do not fit here, and belong above where the study is being justified.

line 83 - this approach should be briefly defined here so that the reader understands what it will entail.

line 84-85 - this sentence should also be moved up in the text to a place before the study is described as part of the justification. 

lines 88-98 - this text is extremely wordy and must be condensed. Most of it belongs in the discussion and conclusion rather than the introduction, as it relates to wider implications of the study.

Methods

line 102 - change ‘for’ to ‘to forecast’

line 104 - ‘technique listed’

line 105 - isn’t the geographic scale of this study larger than the scale at which these methods are used? (i.e. I-O is done for a single country or province).

lines 100-111 - this section can be condensed as a caveat and moved to the discussion

line 113 - ‘on its main host crops’ (change throughout)

line 114 - ‘, the Balkans, and the MENA region’

line 130 - should this be ‘obtained directly from’? Also, this sentence begs the question ‘answers to what’? so a reference to the questionnaire should be made here.

Tables 1-2 can likely be moved to the supplement as they are provided for information purposes rather than to highlight key findings.

line 152 - choose either ‘survey’ or ‘questionnaire’, but it is redundant to write both (I would choose questionnaire because you have an exhaustive set of respondents).

lines 156-160 - rephrase as ‘Weights were assigned to each question using coefficients in relation to how much each respond impacted overall potential risk, where all weights summed to 1’ - but these weights  require justification

Lines 163-165 - rephrase to ‘After re-weighting, total scores could range between 2 and 12’

Line 166 - Remove ‘Therefore’ and provide detail on what was ‘considered carefully’ here

Line 170 - I suggest opening this section with a definition of ‘assumptions scenarios’

Line 185 - are these scores from the questionnaire or from something else? Please clarify

Line 187 - ‘percentiles were rescaled into the adopted scale range’

Line 189 - ‘Later, risk scores were rescaled back into yield loss percentiles for each of the main host crops according to Table 3;

Table 3 can be removed and its caption can be added to Table 4 since the same information is present in both. It can be explained that the top part of table 4 was used as a conversion tool between risk scores and percentiles in general.

The title of Table 4’s middle section should be in units of scale range (1-6), because the percentile conversion is only done in the bottom section. This bottom section should have its own title clarifying it is the ‘associated percentage loss in yield based on scale range score’ or similar.

Section 2.3.3 - I would like to see equations in this section to help understand the methodology used

Lines 203-204 - I would reframe as ‘The socio-economic impacts of the losses of important crops in Table 1 can be grouped into’

Line 206 - ‘is considered as’ —> ‘is proxied by’

Line 207 - ‘We used the yield loss coefficient to calculate potential yield loss’

Line 211 - ‘For olives, grove age in each country (from EFSA) was considered when estimating yield loss’ 

Lines 212-213 - I’m not sure this sentence adds anything and is phrased confusingly - delete?

Line 215 - ‘based on their average age over the period of 2015-2019.

Lines 220-221 - delete - redundant info

Lines 221-223 - “We chose to examine potential impacts to  olive grove distributions in an earlier period (1980-2010), as there was an Xf outbreak in Apulia in 2013.” Were

Lines 223-228 - remove most of this wording and just explain where the data come from and what it is, i.e. ‘ Grape production was separated into wine and table grapes based on data from x’. It is unclear in what way grapes and citrus were ‘a similar case’ to olives - was age structure also taken into account here? Was the threshold also 30 years? And if so, why is 30 years the appropriate age class separator for these species? Please provide a reference. Were all costs estimated form 2015-2019? If so, state this explicitly in this paragraph.

Line 234 - “the problem of missing data’, please indicate the number of missing entries here

Line 232 - convert this paragraph to past tense

Line 238 - ‘has been’ —> ‘was’

Line 243 - ‘it has been elaborated’ — ‘These data were obtained from’

Line 246 - no need to introduce RICA acronym if not used again

Line 247 - Add something like - ’We thus assumed that these Sicilian data were representative of the entire study region’

Line 263 - is it reasonable to assume that there would not be a ban on potentially contaminated crops? Provide a reference if so, otherwise indicate this is an underestimation of impact.

Line 264 - I suggest defining supply disaggregation here for the non-economist reader

Line 273 -274- “Social impact assessment”

Line 280 - “The first was an intensive grove…”

Line 281 - “while the second was an extensive..’

Lines 280-283 - Clearer wording would be “with trees spaced on a x by x m grid” or similar 

Line 288 - does this migration occur following changes in agriculture yield and margin? Clarify here.

Line 290 - “indices” is the plural of index.

Line 298 - what does ‘it’ refer to? The GNI or the entire index? Please clarify

Line 302 - what are ‘the points above’? Please include equations here to understand how expert elicited ratings were transformed into overall scores

Line 305 - Are these the values in Table 4 or something else? Please clarify

Results

Line 309 - ‘European Balkans” —>”Balkans” (for consistency throughout)

Line 314 - “regulatory status of Xf”

Line 320 - be consistent with the wording of ‘vulnerability’ and ‘exposure’ in what these scores are measuring

Line 329 - isn’t Syria’s score 4.35/6 above?

Line 334 -  For ease of interpretation, I suggest indicating with braces, vertical lines, or similar which bars correspond to each region

Sections 3.2 and 3.3 would be clearer in past tense with rephrasing of ‘was estimated at’ vs. ‘is calculated at’,

Section 3.2 is better summarized in a map (ideally, or a table) placed in the main manuscript rather than in text, with only a few sentences written for major results.

Discussion

Lines 415-416 - replace this sentence with the links and implications you are alluding to

Line 418 - wider than what?

Line 424 - I’m not sure what an ’inter-temporal’ horizon is, especially since impacts were only estimated fora  single timepoint

Line 427 - start new paragraph here

Line 434-457 - this should be in past tense because it refers it to previous studies - and should be reframed as a comparison between previous findings and those presented here. Otherwise, the discussion doesn’t put your results in context. What do your results mean for these previous findings?

The conclusions section reads more like an abstract - I would condense it to the main findings rather than detailing the question.

Lines 481-485 - this finding was not emphasized in the results section. It also reads like necessary outcome of how the social vulnerability was calculated and thus a tautology. In other words, if you a priori decide on how to measure social vulnerability and use it as your metric of risk, the only possible outcome is that whichever factors were chosen to represent vulnerability will have high risk. I would replace this wording with something about how this study identifies specific regions at high risk due to high social vulnerability, and then name those regions.

Table A4 - how were these weights decided? how sensitive are the results to these weights?, ‘Establishment and spread’ misspelled. Why are some countries n.a. for social vulnerability? This table would be a great map placed in the main text.

Table A5 - ‘Spain’ misspelled and ‘establishment’ misspelled again

Author Response

Dear reviewer,

Thank you in advance.

Reviewer 2 Report

The authors assess the risk and potential socio-economic impacts of Xylella fastidiosa in the Mediterranean basin. Among countries, risk levels are identified and costs projected according to different crops and activity sectors. Overall, I found this to be a useful study given the paucity of cost assessments for several invasive alien species in this regions. 

Generally, I have only few comments on this manuscript. I found the introduction to be sufficient, methods appropriately explained and justified, and results straightforward to follow. The Discussion, however, is relatively short and would benefit from further development, such as to highlight future work or recommendations for management strategy. At the beginning of the Methods, the economic approaches that were not used would be better placed in the Discussion as well.

The authors may also want to compare the cost magnitude here to that over the entire Mediterranean basin (10.3897/neobiota.67.58926) or Europe (10.3897/neobiota.67.58196), as recently synthesised. Along these lines, I encourage the authors to submit these costs to the InvaCost project, which compiles cost information for invasive alien species worldwide ([email protected]https://doi.org/10.6084/m9.figshare.12668570.v5).

Author Response

Dear Reviewer,

Thank you.

Reviewer 3 Report

The paper of Cardone et al. provides some information about the potential risk posed by the possible introduction of Xylella fastidiosa in countries of the Mediterranean Basin and Balkan area.

The data are mainly obtained from the official public sources of the countries, including the phytosanitary services.

The study offers an idea on the possible negative impact related to the introduction of the pathogen in any single country.

A major concern is posed by the absence in the study of the "climate suitabilty" for the pathogen in a certain area based on real climatic parameters. There are relevant studies on the possible adaptation of Xylella fastidiosa in the Mediterranean Basin that are neither quoted or commented.

Such studies offer a basis for a comparison with the methods applied in the present assessment.

The first Figure in the text is labelled as "Figure 2" even though no Figure 1 is present.

Author Response

Dear Reviewer,

Thank you.

Round 2

Reviewer 3 Report

The manuscript has been improved accordingt o the suggestion raised up.